# Ultrasonographic Visualization of the Ovaries to Detect Ovarian Cancer According to Age, Menopausal Status and Body Type

**DOI:** 10.3390/diagnostics12010128

**Published:** 2022-01-06

**Authors:** Edward J. Pavlik, Emily Brekke, Justin Gorski, Lauren Baldwin-Branch, Rachel Miller, Christopher P. DeSimone, Charles S. Dietrich, Holly S. Gallion, Frederick Rand Ueland, John R. van Nagell

**Affiliations:** 1Division of Gynecologic Oncology, Department of Obstetrics and Gynecology, University of Kentucky Chandler Medical Center-Markey Cancer Center, Lexington, KY 40536-0293, USA; justin.gorski@uky.edu (J.G.); labald1@uky.edu (L.B.-B.); raware00@uky.edu (R.M.); cpdesi00@uky.edu (C.P.D.); charles.dietrich@uky.edu (C.S.D.); Holly.Gallion1@uky.edu (H.S.G.); fuela0@uky.edu (F.R.U.); jrvann2@uky.edu (J.R.v.N.J.); 2Department of Obstetrics and Gynecology, University of Kansas Medical Center, Kansas City, KS 66160, USA; ebrekke@kumc.edu

**Keywords:** transvaginal ultrasound, ovary, age, BMI, menopausal status, visualization, detection, body type, weight

## Abstract

Because the effects of age, menopausal status, weight and body mass index (BMI) on ovarian detectability by transvaginal ultrasound (TVS) have not been established, we determined their contributions to TVS visualization of the ovaries. A total of 29,877 women that had both ovaries visualized on their first exam were followed over 202,639 prospective TVS exams. All images were reviewed by a physician. While visualization of both ovaries decreased with age, one or both ovaries could be visualized in two of every three women over 80 years of age. Around 93% of pre-menopausal women and ~69% of post-menopausal women had both ovaries visualized. Both ovaries were visualized in ~72% of women weighing over 300 lbs. and in ~70% of women with a BMI over 40. Conclusions: Age had the greatest influence on the visualization of the ovaries. The ovaries can be visualized well past the menopause. Body habitus was not limiting to TVS ovarian imaging, and TVS should be considered capable of imaging one or both ovaries in two of every three women over 80 years of age. Thus, older and obese patients remain good candidates for TVS exams.

## 1. Introduction

Transvaginal ultrasonography (TVS) is used as the first-line imaging approach by radiologists, gynecologists and gynecologic oncologists for the evaluation of women suspected to have an adnexal mass that could be a malignant ovarian cancer [1]. TVS requires no preparation, is well-tolerated, is free of radiation, can be completed quickly and yields high quality detailed images of the pelvis [2], including early and late-stage malignancies of the uterus [3] and ovary [4,5,6,7]. TVS has been reported to be significantly more accurate than bimanual clinical examination in detecting the ovaries especially since ovaries frequently are not palpable in women over 55 years of age, or in women weighing more than 200 lbs. [8,9]. Although useful, safe and in wide clinical application, the ultimate limits of modern ultrasonographic instrumentation to detect human ovaries have not been fully characterized. We have reported on ovarian size determined by TVS in a large screening population [10,11], but little is known about how the detectability of the ovaries is influenced by age, body habitus and menopausal status. Importantly, the expectation that ovarian structure may or may not be viewable by TVS can directly impact patient care, especially if imaging is not performed in circumstances when visualization is possible. For example, if a patient is perceived as too old for the ovaries to be visible by TVS, she may only receive a bimanual exam. Consequently, the detection of adnexal pathology may be missed by the physical exam and treatment delayed. However, with an accurate evidence-based perception of the likelihood of ultrasound being able to visualize the ovaries, the opportunity to receive a timely and safe TVS exam will increase the prospects for a clinical intervention if needed.

The objective of this study was to understand how age, menopausal status, weight and BMI influence TVS visualization of the ovaries in order to provide accurate expectations for the likelihood of TVS to successfully visualize these structures.

## 2. Materials and Methods

A total of 46,814 women who enrolled in the University of Kentucky Ovarian Cancer Screening Trial (UKOCST) from January 1987 to June 2018 were evaluated. This prospective cohort trial was approved by the University of Kentucky Institutional Review Board for Human Studies (IRB# 45030) and is registered at ClinicalTrials.gov: NCT04473833. The informed consent was approved institutionally and administered by the study sonographers who were also able to answer questions from participants. Eligibility criteria included (1) all asymptomatic women aged ≥50 years, and (2) asymptomatic women aged ≥25 years with a documented family history of ovarian cancer in at least one primary or secondary relative. All study participants completed a questionnaire that included medical history, surgical history, menopausal status, hormonal use and family history of cancer, as previously published [3]. Menopause was defined when women self-reported that they had not experienced menses for 12 consecutive months. This definition of menopause is not as comprehensive as the World Health Organization definition [12], which includes demonstrated loss of ovarian follicular activity or follicle depletion. The definition used here is as described by the National Institute of Aging [13] and is well-suited for self-reporting. In the self-reporting context, perimenopause was defined as ovulation that is unpredictable with the length of time between periods becoming longer or shorter, or the skipping of some periods or a persistent change of seven days or more in the length of the menstrual cycle [14]. Women with a known ovarian tumor or a personal history of ovarian cancer were excluded from the present investigation.

Transvaginal ultrasonography and color Doppler were performed using General Electric Voluson P5, P8 and P10 units with a 4 to 11 mHz vaginal probe on women with an empty bladder. In performing the TVS exam, the transducer was gradually inserted while observing the ultrasound image on a monitor. The urinary bladder was used as a consistent landmark in the pelvis relative to much more variable positions of the uterus and the ovaries for assessing the orientation of the transducer. Three scanning approaches were used to comprehensively assess the pelvis:(a)Aide-to-side movements to achieve sagittal imaging,(b)90° rotation to obtain semi-coronal images and angulation of the probe vertically,(c)Varying the depth of probe insertion to expose different pelvic structures within the field of view.

The pelvis was surveyed by slowly sweeping the beam in a sagittal plane from the midline to the lateral pelvic side walls followed by turning the probe 90 degrees into the coronal plane and sweeping the beam from cervix to the fundus. Landmarks for proving structure consisted of identifying the iliac vessels in the pelvic sidewall and the tubal vessels located posterior and parallel to the fallopian tubes. Pressure was applied to at least three regions of the abdominal surface to achieve bowel repositioning in order to assist visualizations. All images were reviewed by a physician and by at least one of the authors. The study protocol specified that ovaries be measured in three dimensions. Ovarian volume was calculated using the prolate ellipsoid formula (length × width × height × 0.523) [15,16]. Thus, a visualization event was validated by findings that obtained all three measurements. All screening information was entered into a database (MEDLOG Systems, Crystal Bay, NV, USA) on a local network. Women who had a normal screen were scheduled to return in 12 months for a repeat screen. Only women with two visible ovaries on their first TVS encounter were utilized in this study and then followed over the course of annual examinations by TVS. Women who underwent abdominal surgery were censored from the present analysis so that surgical interventions did not influence visualization outcomes. Because women with a single visualized ovary are already in a transition to non-visualization, we made the assumption that the transition to the loss of ultrasonographic ovarian visualization should begin when both ovaries were present initially so that the incidence of complete non-visualization could be accurately estimated.

### Statistical Analyses

For all analyses, a two-sided significance level of *p* < 0.05 was considered statistically significant. The rate of an event (*r*) was determined as the proportion in a group with both ovaries visualized (*r* = [N_visualized_/[N_nonvisualized_ + N_visualized_]]). The probability ratio (PR) for each group was determined as the rate of visualizing both ovaries in each age group (R_G_) relative to the rate of visualizing both ovaries in the comparison age group (R_comparison_) where 20–30-year-old women comprised the comparison group (PR = R_G_/R_comparison_). The odds of an event (O) were determined as those in a group with both ovaries visualized relative to those where both ovaries were not visualized (O = [N_visualized_/[N_nonvisualized_]]). The odds ratio (OR) for each group was determined as the odds of visualizing both ovaries in an age group (O_G_) relative to the odds of visualizing both ovaries in the comparison group (O_comparison_) where 20–30-year-old women comprised the comparison group (OR = O_G_/O_comparison_). Rates, probability ratio, odds, odds ratio, log odds, Phi coefficient of association, Chi-square test of association with Yates and Pearson *p*-values, Fisher exact probability test, multiple regression analysis and confidence intervals were obtained using VassarStats based on logistic regression [17].

## 3. Results

In total, 29,877 women were identified that had both ovaries visualized on their first TVS exam and were subsequently followed over a course of 202,639 prospective TVS exams. Their demographic characteristics are presented in Table 1.

Age—Visualization of both ovaries decreased with age, but one or both ovaries could be visualized in two of every three women over 80 years of age, while 50% of patients aged 77–78 had both ovaries identified (Figure 1 dashed line). The line of best fit for visualization of both ovaries is shown in green (5th degree polynomial, (*r* = 0.9992), y = 102.097937 − 5.571464X + 2.739043X^2^ − 0.626314X^3^ + 0.048908X^4^ − 0.001299X^5^). The 5th degree polynomial line of best fit for visualization of neither ovary (red line) was y = −0.065979 + 0.719492X − 0.485964X^2^ + 0.113695X^3^ − 0.005843X^4^ + 0.000076X^5^ (*r* = 0.9996). A crossover point was noted for women in their mid-80s, where non-visualization of both ovaries was greater than visualization. The profile for visualization of neither ovary increased after age 50 (Figure 1, red line) to a maximum after age 85 of 35%. After age 40, both the probability ratio of visualizing both ovaries and the odds ratio were significantly different (*p* < 0.001), Table 2.

Body Habitus—Neither weight (Figure 2A, *p* > 0.2) nor BMI (Figure 2B, *p* > 0.12) was independently associated with ovarian visualization since visualization of both ovaries was ~70% across all weights and BMIs. Adjustable probe frequency allowed both ovaries to be visualized in ~72% of women that weighed over 300 lbs. (Figure 2A) and in ~70% of women with a BMI over 40. No more than a 4% difference occurred over the range of 100–300+ lbs. (50–130 kg) or BMIs of 16–40+ (Figure 2B).

Menopausal Status—Menopausal status was self-reported. Both ovaries were visualized in ~93% of pre-menopausal women and ~69% of post-menopausal women (Figure 2C, *p* < 0.001).

Multiple regression analysis by the direct method did not support a model where all variables (R^2^ = 0.09347) significantly contributed to predicting visualization of both ovaries. In this model, each unit change in age was the most significant predictor (*p* < 0.0001), while change in BMI was 8% of age, menopausal status was 5% of age and weight was 4% of age, supporting the data shown in Figure 1 and Figure 2.

## 4. Discussion

TVS should be considered capable of imaging one or both ovaries in two of every three women over 80 years of age. Weight and BMI had little effect on ovarian visualization. Likewise, body habitus was not an independent factor limiting ovarian imaging. Menopause should not be interpreted as an absolute indicator that limits sonographic ovarian visualization, although it is related to age. The importance of the present report is that neither body habitus nor menopausal status was found to significantly impair the ability of TVS to visualize the ovaries.

### 4.1. Clinical Implications

Age-related estimates of ovarian visualization described here indicate that the application of TVS imaging should be considered viable for elderly women so that age should not be used to deny access to TVS. With modern ultrasonographic instrumentation, age should not deter a physician’s decision for sonographic exploration of the ovaries. Importantly, the opportunity to visualize the ovaries is also viable across the full range of weights and BMIs. Obesity and extreme obesity have become more prevalent in American women, increasing from 10% in 1979–1980 to 40+% in 2013–2014 [18]. This increase has been suggested to present a mounting challenge to obtaining high quality TVS images of the abdomen [19]. Abdominal ultrasound is widely considered the radiologic approach most affected by obesity because the depth of insonation needed in very large women attenuates the ultrasound beam. However, TVS reduces the distance between the vaginal transducer and pelvic structures allowing detailed visualization of the ovaries. TVS lacks ionizing radiation and is not subject to absolute weight/girth restrictions as are CTs or MRIs. TVS provides an opportunity to bypass the abdominal apron of excessive skin and fat that characterizes obesity. Glanc et al. have reported that there is a paucity of evidence-based literature on the implications of obesity on TVS [15]. Here we report that modern TVS instrumentation is capable of achieving equivalent ovarian visualization in both obese and non-obese women. Because of the substantial number of ultrasound cases in the current study, the findings reported here can be confidently relied upon in the clinical setting.

### 4.2. Research Implications

The results reported here are prospective findings gleaned from long-term data collection in a large ovarian screening study. While it could be possible that newer instrumentation employed over the 30-year course of data collection might influence visualization; however, there was only a 2.6% increase in the visualization of both ovaries in the most recent 15 years of TVS over the first 15 years. This estimate relies on having sufficient numbers in the groups used to compare effects due to advances in ultrasound technology. Indeed, in the first five years of the screening program, visualizations of both ovaries were 30.2% (*n* = 2068) less than the most recent 15 year experience (*n* = 24,808), while this difference decreased to 0–6% with each 10 year increment, indicating that the earliest ultrasound instruments in this study performed less well. However, the bias represented by these early screens is limited because they account for only 4.4% of the total first screen observations (2068/46,814). Although admission to the study was weighted so that the number of women at least 50 years of age (85%) was greater than women younger than 50 years of age (15%), a substantial number of measurements (*n* = 30,303) were collected for the younger age group. Future investigations should focus on identifying factors correlated with aging that contribute to failures to visualize the ovaries, including bladder and bowel dysfunction.

### 4.3. Strengths and Limitations

The strengths of this paper arise from the large group size of this prospectively studied cohort of asymptomatic women and from criteria for reporting ovarian visualization involving measurement in three planes. By concentrating on women that had both ovaries visualized on their first encounter, this study avoids any timing bias originating in ovarian events occurring prior to entry into the study. The primary timing bias is minimized by examining the transition to complete non-visualization from when both ovaries are visualized. It is possible that the group studied varies from the symptomatic population seen in the clinic, which may affect generalizability; however, women with abnormal TVS findings are included in this study as they constitute a visualization event. Women whose ovaries were removed were excluded; yet, any bias presented by those that were excluded is small, representing ~2.5% of the women with both ovaries visualized on the initial TVS exam.

## 5. Conclusions

Age had the greatest influence on the visualization of the ovaries. Although menopause is often perceived as associated with aging, both ovaries could be visualized well past the onset of menopause. In the majority of women, the ovaries could be visualized for 20–25 years beyond menopause which occurs on average at age 50 [20,21,22]. The present paper provides evidence that the evaluation of ovarian structure using TVS is possible at almost any age and weight. Thus, older and obese patients remain good candidates for TVS exams, providing solution to the limitations that occur with physical examination [8,9].

## Figures and Tables

**Figure 1 diagnostics-12-00128-f001:**
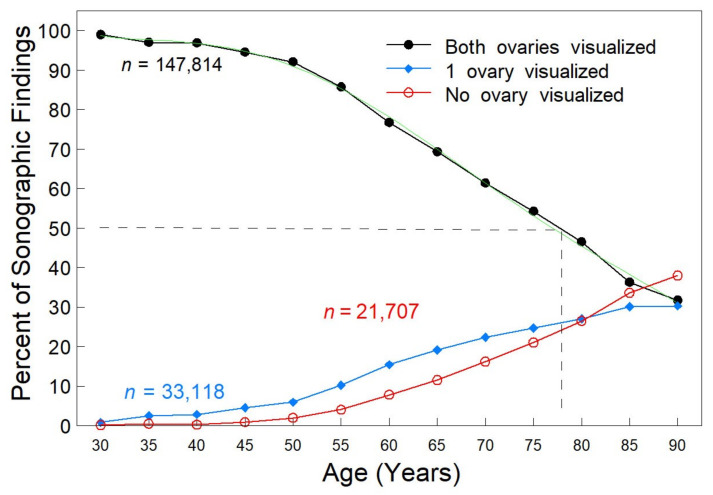
Ovarian visualization relative to age. Sonographic findings visualizing both ovaries (black line), one ovary (blue line) or neither ovary (red line). Age was self-reported and associated with 202,639 TVS encounters. Percentages were determined within each age group. The profile for the visualization of both ovaries was very well fitted by a 5th degree polynomial (Figure 1, green line, *r* = 0.9992) for the probability of visualization over the range 25–90 years of age. *n* = number of TVS observations.

**Figure 2 diagnostics-12-00128-f002:**
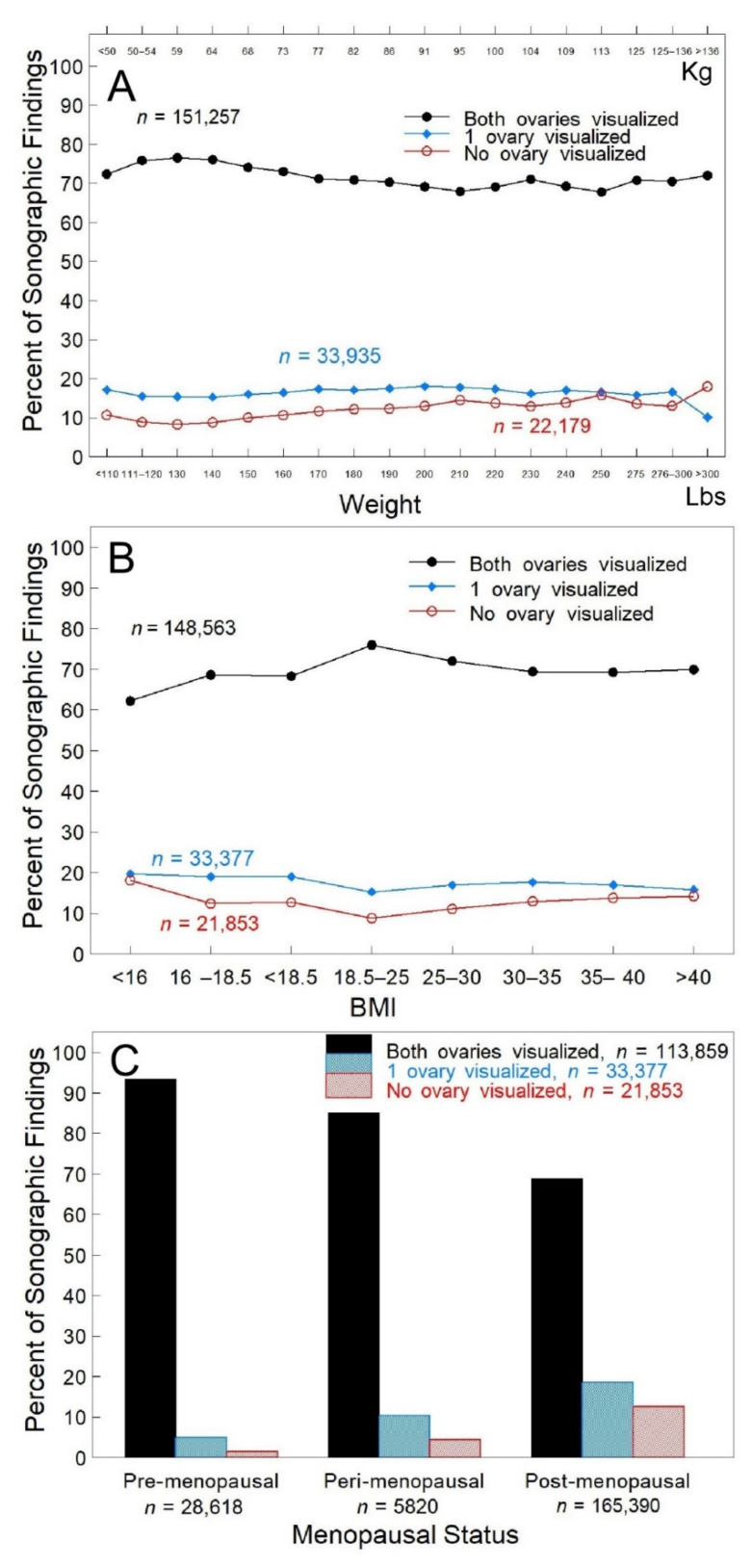
Ovarian visualization relative to weight, BMI and menopausal status. Panel (**A**): Weight was reported in 207,371 TVS encounters. Panel (**B**): Weight and height were reported for the calculation of BMI in 213,793 TVS encounters. Panel (**C**): Menopausal status was self-reported at the time of the TVS exam in 199,828 TVS encounters. Sonographic findings visualizing both ovaries (black bar), one ovary (blue bar) or neither ovary (red bar). Differences in the number of TVS exams reflect unreported events (missing data) resulting in exclusion from analysis. Percentages were determined for each grouping.

**Table 1 diagnostics-12-00128-t001:** Demographic characteristics of women undergoing TVS. Data represented as mean, median, range in parentheses. Subject data are for first encounter, while encounter data are across all encounters.

	All Subjects (*n* = 29,877)	All Encounters (*n* = 202,639)
Age (y)	55.0, 55 (20–91)	60.1, 60 (20–95)
Weight (kg)	73, 70.3 (38–204)	72.3, 69.4 (36–205)
Height (cm)	163.5, 162.6 (119–198)	163.6, 162.6 (119–198)
BMI	27.3, 26 (13–80)	27, 26 (13–80)
Pre-menopausal Peri-menopausal Post-menopausal	5966 (20.9%) 1262 (4.4%) 21,251 (74.6%)	28,618 (14.3%) 5820 (2.9%) 165,390 (82.8%)

**Table 2 diagnostics-12-00128-t002:** Probability and odds ratios of ovarian visualization by age. The 95% confidence interval is in parentheses.

Age (yrs)	Total *n*	Both Ovaries Visualized	PR [95% CI]	OR [95% CI]
20–30	1182	1170	1	1
31–35	2439	2366	0.98 (0.9712–0.9889)	0.3324 (0.1798–0.6146)
36–40	5103	4943	0.9786 (0.9712–0.9860)	0.3169 (0.1756–0.5717)
41–45	8483	8019	0.955 (0.9477–0.9624)	0.1773 (0.0996–0.3154)
46–50	13,096	12,056	0.93 (0.9229–0.9372)	0.1189 (0.0671–0.2107)
51–55	33,202	28,463	0.8661 (0.8598–0.8724)	0.0616 (0.0349–0.1089]
56–60	39,278	30,132	0.775 (0.7689–0.7812)	0.0338 (0.0191–0.0597)
61–65	38,077	26,396	0.7003 (0.6942–0.7065)	0.0232 (0.0131–0.0409)
66–70	30,164	18,522	0.6203 (0.6138–0.6270)	0.0163 (0.0092–0.0288)
71–75	18,823	10,210	0.548 (0.5402–0.5559)	0.0122 (0.0069–0.0215)
76–80	9077	4221	0.4698 (0.4592–0.4806)	0.0089 (0.0050–0.0158)
81–85	3009	1092	0.3666 (0.3496–0.3845)	0.0058 (0.0033–0.0104)
>85	706	224	0.3205 (0.2876–0.3572)	0.0048 (0.0026–0.0086)

## Data Availability

De-identified data sharing is available after institutional approval by the University of Kentucky and the requestor’s institution/employer through Institutional Review Board approved protocols at each.

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
