# Peer review of "Ultrasonographic Visualization of the Ovaries to Detect Ovarian Cancer According to Age, Menopausal Status and Body Type"

_diagnostics, 2022, doi:10.3390/diagnostics12010128_

Round 1

Reviewer 1 Report

The manuscript sent for review concerns a very interesting topic of ovarian imaging depending on age, menopausal status and BMI. This is an important problem for which there are not many scientific studies, and it is particularly important in the situation of refusal to perform the ultrasound examination due to age or obesity of the patient. The authors clearly point to this. A particular value of the research is the large number of patients and ultrasound examinations performed.

The manuscript is well written and suitable for publication with a few corrections and additions.

My comments:

  1. There is no reference to ovarian cancer except in the title of this work. Visualizing the ovaries is one thing, and assessing a tumor for malignancy is another.
  2. In the summary, I believe that all statistical methods are unnecessarily mentioned.
  3. The conclusions in the summary are not in fact conclusions, they are results. This should be changed. Better conclusions appear at the end of the manuscript and differ from those in the abstract.
  4. The research covers a period of over 30 years. Could the progress in ovarian visualization with technological advances be assessed? There is only a slight mention in the manuscript compared to the two fifteen years periods. This requires more discussion.
  5. The authors misdefine menopause. Please refer to the WHO definition.
  6. Moreover, the concept of perimenopause appears in some results and tables. What does it mean? No explanation.
  7. Why were only women who had both ovaries visualized at the first examination included in the study? The explanation in the discussion is unconvincing.
  8. The body weight of women is given in lbs and kg, this requires standardization.
  9. The description of Figure 1. contains a lot of information which is the result of research. Why is this in the caption under the figure?
  10. Figure 2. What is peri-menopause?
  11. A good conclusion is the last sentence of the first paragraph of the discussion.

The manuscript is ready for publication after the above additions and corrections have been made.

Author Response

Thank you for your detailed review. We have responded with edited changes to the manuscript for all your comments.  Line numbers that are referred to correspond to “track changes on” showing both additions and deletions.

Reviewer #1

The manuscript sent for review concerns a very interesting topic of ovarian imaging depending on age, menopausal status and BMI. This is an important problem for which there are not many scientific studies, and it is particularly important in the situation of refusal to perform the ultrasound examination due to age or obesity of the patient. The authors clearly point to this. A particular value of the research is the large number of patients and ultrasound examinations performed.

The manuscript is well written and suitable for publication with a few corrections and additions.

My comments:

1. There is no reference to ovarian cancer except in the title of this work. Visualizing the ovaries is one thing, and assessing a tumor for malignancy is another.

Reference to ovary cancer has been added at the beginning of the introduction in lines 39-43.

2. In the summary, I believe that all statistical methods are unnecessarily mentioned.

Statistical methods have been removed from the Abstract. See track changes lines 18-31

3. The conclusions in the summary are not in fact conclusions, they are results. This should be changed. Better conclusions appear at the end of the manuscript and differ from those in the abstract.  In line 31-34 we have adjusted the conclusions in the summary so that conclusions that the referee identifies are included and have removed results.

4. The research covers a period of over 30 years. Could the progress in ovarian visualization with technological advances be assessed? There is only a slight mention in the manuscript compared to the two fifteen years periods. This requires more discussion. More discussion has been added at lines 221-230, specifically focusing on the very earliest data collecting years.

5. The authors misdefine menopause. Please refer to the WHO definition.

At lines 74-82 we address this point, defining menopause in terms of self-reporting and the National Institute of Aging definition.

National Institute of Aging https://www.nia.nih.gov/health/what-menopause

WHO definition:  David G. Weismiller. Menopause, Primary Care: Clinics in Office Practice, Volume 36, Issue 1,2009, Pages 199-226, ISSN 0095-4543, https://doi.org/10.1016/j.pop.2008.10.007.    The World Health Organization (WHO) defines natural menopause as the permanent cessation of menstruation resulting from the loss of ovarian follicular activity or follicle depletion. Natural menopause is recognized to have occurred after 12 consecutive months of amenorrhea for which there is no other pathologic or physiologic cause. Menopause occurs with the final menstrual period, which is known with certainty only in retrospect a year or more after the event.An adequate biologic marker for menopause does not exist. Documentation of amenorrhea for at least 1 year with an associated serum follicle-stimulating hormone (FSH) level of greater than 50 IU/mL and a circulating serum estradiol level of less than 50 pg/mL have been used

6. Moreover, the concept of perimenopause appears in some results and tables. What does it mean? No explanation.

We explain the perimenopause in edits at lines 78-82.

            Mayo Clinic definition https://www.mayoclinic.org/diseases-            conditions/perimenopause/symptoms-causes/syc-20354666

7. Why were only women who had both ovaries visualized at the first examination included in the study? The explanation in the discussion is unconvincing.

            We have added an explanation at lines 111-114 & 242-244.

8. The body weight of women is given in lbs and kg, this requires standardization.

            We present the data in both measurement units so that the the figures could be interpreted       internationally by users of either lbs or kg. We have made edits at line 173.

9. The description of Figure 1. contains a lot of information which is the result of research. Why is this in the caption under the figure?

            The information referred to has been moved to the results section in lines 140-149.

 10. Figure 2. What is peri-menopause?

We explain the perimenopause in edits at lines 78-82.

We defined the perimenopause as menstrual irregularity prior to achieving 12 consecutive months without a menstrual period.

11. A good conclusion is the last sentence of the first paragraph of the discussion.

            This conclusion has been added to the last line of the Abstract.

The manuscript is ready for publication after the above additions and corrections have been made.

Reviewer 2 Report

Dear authors , 

It is an interesting article however it can be improved in some topics:

Introduction: it can be more extensive . Explaining with more detail who and when is used this method.

Methods: it can have an paragraph referring the informed concern and the ethical approval .

Discussion: must be improved with more recent articles comparing with the results always into account the aims of the study .

References: more references are needed and more recents .

Best regards

Author Response

Thank you for your detailed review. We have responded with edited changes to the manuscript for all your comments.  Line numbers that are referred to correspond to “track changes on” showing both additions and deletions.

Reviwer 2

Dear authors , 

It is an interesting article however it can be improved in some topics:

Introduction: it can be more extensive . Explaining with more detail who and when is used this method.
Edits have been made in the 1st line of the introduction to address this point.

Methods: it can have an paragraph referring the informed concern and the ethical approval .
This has been added and can be found at lines 67-69.

Discussion: must be improved with more recent articles comparing with the results always into account the aims of the study.
We have improved the Discussion section with edits at lines 221-230, 242-244.

References: more references are needed and more recents .
References have been added including four references from 2021